# Simultaneous Detection of Four Main Foodborne Pathogens in Ready-to-Eat Food by Using a Simple and Rapid Multiplex PCR (mPCR) Assay

**DOI:** 10.3390/ijerph19031031

**Published:** 2022-01-18

**Authors:** Aya Boukharouba, Ana González, Miguel García-Ferrús, María Antonia Ferrús, Salut Botella

**Affiliations:** Department of Biotechnology, Centro Avanzado de Microbiología de Alimentos, Universitat Politècnica de València, Camino de Vera, s/n, 46022 Valencia, Spain; petiteyouya24@hotmail.fr (A.B.); angonpel@upv.es (A.G.); migarfe1@upv.es (M.G.-F.)

**Keywords:** food safety, detection, multiplex PCR (mPCR), co-enrichment, organic food, ready-to-eat food, *Escherichia coli*, *Listeria monocytogenes*, *Staphylococcus aureus*, *Salmonella enterica*

## Abstract

The increasing consumption of organic or ready-to-eat food may cause serious foodborne disease outbreaks. Developing microbiological culture for detection of food-borne pathogens is time-consuming, expensive, and laborious. Thus, alternative methods such as polymerase chain reaction (PCR) are usually employed for outbreaks investigation. In this work, we aimed to develop a rapid and simple protocol for the simultaneous detection of *Escherichia coli* (*E coli*), *Listeria monocytogenes* (*L. monocytogenes*), *Staphylococcus aureus* (*S. aureus*) and *Salmonella enterica* (*S. enterica*), by the combination of an enrichment step in a single culture broth and a multiplex PCR (mPCR) assay. The effectiveness of several enrichment media was assessed by culture and PCR. Buffered peptone water (BPW) was selected as the optimum one. Then, mPCR conditions were optimized and applied both to pure co-cultures and artificially inoculated food samples (organic lettuce and minced meat). In the culture medium inoculated at 10^0^ CFU/mL, mPCR was able to detect the four microorganisms. When performed on artificially food samples, the mPCR assy was able to detect *E. coli*, *S. enterica*, and *L. monocytogenes*. In conclusion, BPW broth can effectively support the simultaneous growth of *E. coli*, *S. aureus*, *L. monocytogenes*, and *S. enterica* and could be, thus, used prior to a mPCR detection assay in ready-to-eat food, thereby considerably reducing the time, efforts and costs of analyzes.

## 1. Introduction

The development of food industrialization and the international trade in fresh and frozen foods are the two main factors that have contributed to the increasing complexity of the global food security challenge [1]. However, the resurgence of certain food trends such as massive consumption of organic food, the raw food diet, and ready-to-eat food have resulted in an alarming increase of food poisoning cases [2,3]. Despite the health benefits, the raw food diet, which is defined by the consumption of raw vegetables or animal products, when not subjected to heat treatment over 40 °C, may be the cause of serious foodborne illnesses and even disease outbreaks. Among the outbreaks reported by the Centers for Disease Control (CDC) and Prevention’s Foodborne Disease Outbreak Surveillance System (FDOSS), 18 were identified involving vegetables and dairy products from 1992 to 2014, resulting in 779 illnesses, 258 hospitalizations, and 3 deaths [4].

Among the foodborne outbreaks that represent a cause for concern, most come from animal origin food, including beef meat, poultry, eggs, raw milk and milk products [5,6,7,8], and vegetables such as radishes, cucumbers, carrots, leafy foods such as lettuce, spinach, sprouts, cabbage, or even fruits [9,10,11,12,13,14,15], which may be contaminated by multiple pathogens such as *Listeria monocytogenes* (*L. monocytogenes*), *Staphylococcus aureus* (*S. aureus*), *Salmonella enterica* (*S. enterica*), and *Escherichia coli* (*E. coli*) [10,15,16]. Those microorganisms are defined as significant foodborne pathogen due to the severity of diseases and the number of illness cases caused [15,17,18,19]. To guarantee food safety, obtain the possibility of locating potential outbreaks, and limit the spreading of infection, the screening for these pathogens needs to be carried out on raw food products prior to and throughout the complete distribution process [20,21].

The current detection methods based on microbiological cultures require pre-enrichment and selective enrichment steps, followed by isolation on several selective media and biochemical or serological tests to obtain a definitive result [7,22]. These procedures are still considered as the “gold Standard”, although they are complicated, extremely labor intensive, costly, time consuming (from days to weeks), and might be subject to handling errors. The recent development of molecular methods such as polymerase chain reaction (PCR) has shown a high potential in foodborne outbreak investigation, food analysis and food monitoring [14,23,24,25,26]. Unfortunately, the sensitivity and reliability of PCR methods depend on the number of target bacterial cells. Thus, low contamination levels in food samples make the detection of target pathogens very difficult [7]. Consequently, an enrichment step prior to DNA extraction is generally required to increase the target pathogen concentration in the sample and resuscitate the stressed or sub-lethally injured cells, which improves the detection limits of the PCR and avoids false negative results [7,11,27,28]. To simplify the work required, save time, and reduce costs and humans errors, several studies have attempted to build protocols for the simultaneous detection of multiple pathogens in a single reaction [29,30,31,32]. Most of the recent works are based on the development of a selective co-enrichment broth, to ensure a simultaneous growth of target pathogens and suppress the background microbiota, prior to the detection by multiplex PCR (mPCR). Unfortunately, the presence of several selective agents may have serious repercussions on the growth balance between the different target pathogens, which can favor one target and inhibit the growth or delay the recovery of the others at the same time [19,33]. In addition, most of those synthetic media developed to be selective, are not specific to the target pathogens [5,11,30,33,34,35].

In the present work, we aimed to develop a rapid and simple protocol for the simultaneous detection of four foodborne pathogens *E. coli*, *L. monocytogenes*, *S. aureus*, and *S. enterica*, which could be used for the analyses of ready-to-eat food, by the combination of a co-culture step in a single culture broth and mPCR detection. Buffered peptone water (BPW) broth was selected after comparing its performances with multiple selective and non-selective culture media, as well as analyzing its recovery capacities from low initial inoculums, in individual pure cultures, in co-culture, and in the presence of artificially inoculated food matrices with background microbiota. The construction of our mPCR was based on the combination of four highly selective primers, mentioned in several previous works and optimized to ensure the most sensitive detection possible. Finally, the potential application of BPW as basic co-culture broth before the mPCR, for the simultaneous detection *E. coli*, *L. monocytogenes*, *S. aureus*, and *S. enterica* from artificially inoculated raw food was evaluated.

## 2. Materials and Methods

### 2.1. Bacterial Strains

In this work, the four most relevant pathogenic bacteria in food have been studied. The strains used, *E. coli* CECT 101, *S. enterica* CECT 4266, *L. monocytogenes* CECT 936, and *S. aureus* CECT 435, were procured from the Spanish Type Culture Collection (CECT, Valencia, Spain).

For the evaluation of the mPCR specificity, strains from different origins were used: *E. coli* CECT 425, *E. coli* CECT 418, *E. coli* CECT 4558, *Citrobacter freundii* (*C. freundii*) CECT 401, *Micrococcus luteus* (*M. luteus*) CECT 245, *Staphylococcus epidermidis* (*S. epidermidis*) CECT 231 and 3 laboratory isolates: *Listeria innocua* (*L. innocua*)*, Listeria grayi* (*L. grayi*) and *Bacillus cereus* (*B. cereus*).

Cultures started from frozen stocks preserved at −20 °C, by streaking inoculation on Plate Count Agar (PCA-Scharlau), the different bacterial strains were separately cultivated and the plates were incubated 24 h at 37 °C.

### 2.2. Selection of a Co-Culture Medium

To select the most suitable co-culture broth for the four target bacteria, the effect of several selective and non-selective liquid media (commercial or synthetic) on the growth of *E. coli*, *S. enterica*, *L. monocytogenes*, and *S. aureus* were compared, according to the maximum population rates obtained under the same culture conditions (24 h at 37 °C).

#### 2.2.1. Effect of Various Culture Broths on Individual Growth

The effect of multiple commercial non-selective broths, Luria-Bertani broth (LB) (Difco^TM^-BD-France), nutrient broth (NB) (Difco™-BD-France) and buffered peptone water (BPW) (Scharlau-Spain), on the individual culture of each pathogen was evaluated.

In the same way, the commercial selective media recommended by the ISO and UNE-EN ISO protocols, as specific selective enrichment for the target microorganisms were tested: Brilliant green bile lactose broth (BGBLB) (Difco^TM^-BD-France) for *E. coli* [36], Rappaport Vassiliadis-R10 Broth (RV) (Difco^TM^-BD-France) for *S. enterica* [37], Giolitti-Cantoni (GC) broth (Scharlau-Spain) supplemented with the 06-011-100 potassium tellurite solution 3.5% (Scharlau-Spain) for *S. aureus* [38] and the Fraser enrichment broth base (FB) (Scharlau-Spain) supplemented with *Listeria* UVMII Selective Supplement 06-111-LY01 (Scharlau-Spain) and ferric ammonium citrate supplement 06-112-LY01 (Scharlau-Spain) for *L. monocytogenes* [39]. Moreover, the selective synthetic broth (SSSLE broth), developed by Chen et al. (2015) [5] for the simultaneous growth of *S. enterica*, *S. aureus*, *S. flexneri*, *L. monocytogenes* and *E. coli*, was tested in our work.

To evaluate the growth rates of each microorganism during individual cultures in several broths, one isolated colony of each reference strains cultivated 24 h on PCA, was separately inoculated in each evaluated broth, for a final volume of 10 mL. Then, inoculated broths were incubated during 24 h, at 37 °C for BN, BPW, LB, BGBLB, GC, FB, SSSLE and at 41 °C for RV.

The maximum population rates obtained after incubation from each broth were determined by count, on the respective selective media described in the UNE-EN ISO protocols: tryptone bile glucuronic agar (TBX) (Scharlau-Spain) for *E. coli*, xylose-lysine-deoxycholate agar (XLD) (Scharlau-Spain) for *S. enterica*, Baird Parker agar base (BPA) (Scharlau-Spain) mixed with the egg yolk emulsion potassium tellurite (EYEPT) 06-026-100 (Scharlau-Spain) for *S. aureus* and Palcam agar base (PAL) (Scharlau-Spain), mixed with selective supplement 06-110LY01 (Scharlau-Spain) for *L. monocytogenes*.

All plates were incubated for 24–48 h at 37 °C. The number of colony forming units/mL (CFU/mL) was determined according to the formula mentioned in UNE-EN ISO 7218:2008 [40].

#### 2.2.2. Evaluation of BPW Recovery Capacity

##### Preparation of the Inoculum

Fresh pure cultures were prepared by inoculating separately one isolated colony from each target bacteria in 10 mL BPW, then incubated 24 h at 37 °C under 150 rpm agitation. The final concentration ranging 10^8^–10^9^ CFU/mL was determined in accordance with UNE-EN ISO 7218:2008 [40].

##### Effect of BPW on Individual and Co-Culture Growth

To evaluate BPW recovery capabilities, from low initial inoculum concentrations in individual pure cultures, three initial inoculum levels were tested: 10^3^; 10^2^; 10^1^ CFU/mL, in a final volume of 10 mL, for each target pathogen. The concentration of the four target bacteria was equal in each experiment: Experiment I: (10^3^; 10^3^; 10^3^; 10^3^ CFU/mL); Experiment II: (10^2^; 10^2^; 10^2^; 10^2^ CFU/mL); and Experiment III: (10^1^; 10^1^; 10^1^; 10^1^ CFU/mL).

All plates were incubated on the selective media previously mentioned for 24 h at 37 °C and the number of CFU/mL was determined according to the formula mentioned in UNE-EN ISO 7218:2008 [40]. One mL aliquots from each resulting culture were frozen at −20 °C, to be tested by PCR.

##### Effect of BPW on Co-Culture Growth, from Artificially Inoculated Food Matrix

To evaluate BPW recovery efficiency in co-culture, from artificially inoculated food with background microbiota, two kinds of ready-to-eat food were tested, namely eco-organic lettuce and minced meat, purchased from local stores and used fresh without any disinfectant treatment, in order to preserve the background microbiota of the foods.

Samples (lettuce and minced meat) were handled following the same protocol: 10 g of sample were artificially inoculated with the appropriate pure culture dilutions (prepared in Section 2.2.2.), to obtain 10^3^ CFU/mL initial inoculum concentration of each pathogen, then mixed to 90 mL of sterile BPWin a stomacher bag. Mixtures were homogenized for 5 min and then incubated 24 h at 37 °C.

The number of CFU/mL was determined according to the formula mentioned in UNE-EN ISO 7218:2008 [40]. One mL aliquots from each resulting co-culture were frozen at −20 °C, to be tested by PCR.

### 2.3. PCR Detection

#### 2.3.1. Preparation of DNA Template

DNA templates were extracted by using the GenElute™ Bacterial Genomic DNA Kit (Sigma-Aldrich, St. Louis, MI, USA) with Lysozyme.

To evaluate the thermal lysis extraction protocol effectiveness, without lysozyme, for both Gram-positive and Gram-negative [11], aliquots collected from each individual culture were submitted to DNA extraction by both methods.

##### Thermal Lysis Method

One mL of the collected aliquots was subjected to 10 min centrifugation at 14,000 rpm (≈14,462× *g*), the pellet was resuspended in 100 µL of sterile Milli-Q water, after vortex, the suspension was boiled 10 min at 100 °C and immediately cooled on ice for 5 min. The supernatant obtained after 5 min centrifugation at 12,000 rpm (≈10,625× *g*) was stored at −20 °C, until further use as PCR template.

##### GenElute™ Bacterial Genomic DNA Kit

One mL of the collected aliquots was subjected to 2 min centrifugation at 16,000× *g* and the pelleted cells were used for DNA extraction following Gram-positive manufacturer’s extraction protocol of GenElute™ Bacterial Genomic DNA Kit (Sigma-Aldrich), which requires the use of Lysozyme (Lysozyme BioChemica-PanReac AppliChem). The eluted DNA was stored at −20 °C to be used as PCR template.

#### 2.3.2. Primers

Multiplex PCR (mPCR) was carried out with the combination of four specific pairs of primers in one reaction (Table 1). For the detection of *E. coli,* primers GADA 670-F/R targeting the glutamate decarboxylase enzyme encoding gene (*gadA*), were chosen from McDaniels et al. (1996) [41] study. Nuc 484-F/R primers, targeting the encoding gene of *S. aureus* thermostable nuclease (*nuc*) were selected from the work of Xu et al. (2006) [42]. LM 404-F/R primers from Wu et al. (2004) [43] targeting the listeriolysin gene (*lisA*), were used for the detection of *L. monocytogenes* and the primers SalinvA 284-139/141 from the work of Rahn et al. (1992) [44], targeting the encoding gene of the Invasion protein A (*invA*), were used for the detection of *S. enterica*. The specificity of each primer was checked in silico by using the Primer-Blast program, Basic Local Alignment Search Tool, http://blast.ncbi.nlm.nih.gov/Blast.cgi (accessed on 25 November 2021), to avoid nonspecific products [45]. Then, the primers were synthesized by TIB MOLBIOL (Syntheselabor GmbH-Germany).

#### 2.3.3. Simplex PCR

Primer validation was made individually by simplex PCR, under the same conditions. These conditions were obtained after optimization tests, including an annealing temperature gradient (55; 56; 57; 58; 60 °C), for each simplex PCR.

PCRs were performed according to an initial denaturation at 94 °C for 2 min, 35 cycles of denaturation at 94 °C for 30 s, annealing at 58 °C for 30 s, extension at 72 °C for 60 s and a final extension at 72 °C for 7 min.

Reaction mixtures had similar composition: 2.9 µL Reaction Buffer 10 × NH_4_; 3 mM MgCl_2_ Solution; 0.20 mM dNTPs (dNTP Mix-BIOLINE) and 2.5 Units Taq DNA polymerase (BIOTAQ ™ DNA Polymerase-BIOLINE), mixed with 4 µL of target DNA. The amount of primers used for each simplex was 0.4 µM GADA670 for *E. coli*, 0.4 µM Nuc484 for *S. aureus*, 0.4 µM LM404 for *L. monocytogenes* and 0.2 µM SalinvA284 for *S. enterica*. Reaction mixtures with Milli-Q distilled water, instead of DNA template, were used as a negative control.

After amplification, 5 μL of each PCR products were mixed with Ready-to-Load (GeneRuler-Thermo Scientific) and submitted to electrophoresis for 80 min at 80 V on 1.5% agarose gel (Agarose D1 low EEO-CONDA/TAE buffer PanReac AppliChem, 0.05 M EDTA-Na_2_·2H_2_O, 1 M Acetic Acid glacial, 2 M Tris), then visualized under a UV transilluminator.

#### 2.3.4. Multiplex PCR (mPCR)

mPCR was developed for the simultaneous detection of *E. coli*, *S. aureus*, *L. monocytogenes* and *S. enterica*, in a single reaction [46]. After multiplexing, several optimization tests were carried out including testing of annealing temperature gradient (58; 59; 60; 61; 62 °C), number of cycles variations (30; 25; 20 cycles), amount of MgCl_2_ (3.2; 3.5; 4 mM), dNTPs (0.22; 0.3; 0.4 mM), and the balance between primers.

The best conditions were: A total volume of 29 µL containing at least 100 ng of extracted DNA, 2.9 µL Reaction Buffer 10 × NH_4_; 3.5 mM MgCl_2_ Solution; 2.5 Units Taq DNA polymerase (BIOTAQ™ DNA Polymerase-BIOLINE) and 0.22 mM dNTP’s (dNTP Mix-BIOLINE). The primers concentrations were 0.2 µM GADA670, 1 µM Nuc484, 0.16 µM LM404, and 0.46 µM SalinvA284.

Reaction mixture with Milli-Q water was used as a negative control. The amplification was carried out using a Thermal Cycler (Eppendorf AG-Germany) with an initial denaturation at 94 °C for 2 min; followed by 35 cycles of denaturation at 94 °C for 30 s, annealing at 59 °C for 30 s, extension at 72 °C for 60 s, and a final extension step at 72 °C for 7 min.

After mPCR, 5 μL of PCR products were mixed with Ready-to-Load (GeneRuler-Thermo Scientific) and tested by electrophoresis during 80 min at 80 V on 1.5% agarose gel (Agarose D1 low EEO-CONDA/TAE buffer PanReac AppliChem, 0.05 M EDTA-Na_2_·2H_2_O, 1 M Acetic Acid glacial, 2 M Tris). Amplification products were visualized under a UV transilluminator.

#### 2.3.5. Evaluation of Specificity

The specificity of the mPCR was evaluated by determining the ability of this protocol to distinguish the target bacteria from non-target ones. To check the presence of any cross-hybridization, which could lead to mis-priming between the four primers and four target DNAs, the selectivity of each primer was confirmed by mixing the four pairs of primers with several random combinations of positive control DNAs and then amplified according to the optimized PCR multiplex conditions mentioned above. Extracted DNA from pure cultures of *E. coli* CECT 101, *S. aureus* CECT 435, *L. monocytogenes* CECT 936, and *S. enterica* CECT 4266 were used as positive controls.

Then, *E. coli* CECT 425; *E. coli* CECT 418; *E. coli* CECT 4558, *C. freundii* CECT 401; *M. luteus* CECT 245; *S. epidermidis* CECT 231, and the 3 laboratory isolates *L. innocua*, *L. grayi* and *B. cereus*, were used for the evaluation of the mPCR specificity.

#### 2.3.6. Evaluation of Sensitivity

The sensitivity was evaluated, firstly by using DNAs extracted separately from the four pure cultures in BPW, to determinate detection limits of each simplex and mPCR. Afterwards, tests were carried out by using DNA extracted from co-cultures made in BPW, with and without food matrix artificially inoculated, under background microbiota, as described in Section 2.2.2.

##### Detection Limits of Simplex and mPCR, from Individual Cultures

Detection limits of each simplex PCR were determined by multiple reactions whose composition was similar except for DNA amount, which were tested with a decreasing gradient of DNA amount in a final volume of 29 µL.

The evaluation of the mPCR detection limits was also carried out by several reactions, with the mix of the four specific primers and equal ratios of the four target DNAs, according to a decreasing gradient of DNA amount in a final volume of 29 µL. All of the PCRs were conducted following the optimized conditions mentioned previously.

##### Detection Limits from BPW Co-Culture Recovery

DNA extracted from 1 mL aliquots of each co-culture with the four target microorganisms in BPW broth, initially inoculated at 10^3^; 10^2^; 10^1^; 10^0^ CFU/mL, were tested by mPCR following the optimized conditions mentioned above.

##### Detection Limits from BPW Co-Culture Recovery, with Artificially Inoculated Food Matrices

The detection by mPCR was made by using the DNA extracted from co-culture recovery aliquots from artificially inoculated food matrices (eco-organic lettuce and minced meat) at 10^3^ CFU/mL of each target pathogen, according to optimized conditions mentioned above.

## 3. Results

### 3.1. Selection of a Co-Culture Medium

#### 3.1.1. Effect of Various Culture Broths on Individual Growth

One isolated colony of each reference strains cultivated 24 h on PCA was separately inoculated in each evaluated broth, for a final volume of 10 mL. Then, inoculated broths were incubated during 24 h, at 37 °C. The two non-selective media showing the best recovery performance for the four target bacteria in individual growth without any competition were NB and BPW. *E. coli* growth rates exceeded 10^8^ CFU/mL in all of the non-selective media after 24 h of incubation. For *S. enterica,* growth ranged between 10^7^ and 10^8^ CFU/mL and varied between 10^9^ and 10^10^ CFU/mL for *S. aureus.* Regarding *L. monocytogenes,* the growth rates recorded in all non-selective media were close, approximately 10^8^ CFU/mL.

Maximum population rates of the selective broths recommended for the enrichment step by ISO protocols (*E. coli* in BGBLB, *S. enterica* in RV and *S. aureus* in GC) were much lower than growth rates obtained in the non-selective broths tested, except for FB broth, which got the highest growth rate for *L. monocytogenes*.

The selective co-enrichment medium SSSLE was the least effective of all evaluated media in this study, with the weakest growth rate for *E. coli* and no growth for *S. enterica*, *S. aureus* or *L. monocytogenes* (Table 2).

#### 3.1.2. Effect of BPW on Individual and Co-Culture Growth

The evaluation of BPW recovery capacities from low initial inoculum, in individual pure culture, showed relatively stable and close rates, over 10^8^ CFU/mL for *E. coli*, *S. aureus* and *S. enterica*, and over 10^7^ for *L. monocytogenes* (Table 3).

#### 3.1.3. Effect of BPW on Co-Culture Growth from Artificially Inoculated Food Matrix

Regarding the co-culture in BPW broth from food matrices artificially inoculated at 10^3^ CFU/mL of each target bacteria, the recovery rates from the eco-organic lettuce sample reached 10^8^ CFU/mL for the Gram-negative *E. coli* and *S. enterica*, and 10^6^ CFU/mL for the Gram-positive *S. aureus* and *L. monocytogenes*.

For the artificially inoculated minced meat sample, the recovery rates recorded were lower than those obtained for the lettuce sample, about 10^7^ CFU/mL for the Gram-negative microorganisms and *S. aureus*, while for *L. monocytogenes* it was up to 10^5^ CFU/mL (Table 4).

### 3.2. PCR Detection

#### 3.2.1. Simplex PCR

Simplex PCR detection from individual cultures was applied to check the correct performance of each primer pair and the performance of DNA extraction method. Results showed that the specific amplification of *E. coli*, *S. aureus*, *L. monocytogenes*, and *S. enterica* produced amplicons of different sizes, corresponding to 670 bp, 484 bp, 404 bp, and 284 bp, respectively, appearing as distinct bands on electrophoresis gel, without any non-specific product.

A difference in bands intensity for *L. monocytogenes* and *S. enterica* was recorded for the Thermal lysis extraction protocol. By using the GenElute ™ bacterial genomic DNA Extraction kit with lysozyme, intensity of all bands was much higher, without difference between the four bacteria (Figure 1). According to this result, the kit was used to extract the DNA of the samples for all of the remaining assays.

#### 3.2.2. mPCR

After the primers were validated in silico and by simplex PCR, multiplexing was carried out by a progressive integration of primers, in several duplex and triplex PCR reactions, then the four primers pairs were mixed in a single reaction with their respective target DNAs (quadruplex PCR). To ensure the amplification of the four target fragments and to avoid non-specific reactions, conditions such as the annealing temperature and concentration balance of the four primers in the reaction, were optimized.

The detection could be made correctly in simplex, duplex, triplex, and quadruplex PCR, without any non-specific product visible on the electrophoresis gel.

#### 3.2.3. Evaluation of Specificity

Regarding the evaluation of multiplex detection specificity, for DNAs extracted from reference strains (*E. coli* 101 CECT, *S. aureus* 435 CECT, *L. monocytogenes* 936 CECT, *S. enterica* 4266 CECT), each primer amplified exclusively its own target gene.

On the other hand, for the reference strains *C. freundii* 401 CECT, *M. luteus* 245 CECT and *S. epidermidis* 231 CECT or the three laboratory isolate strains *B. cereus*, *L. innocua* and *L. grayi* no detection was recorded, confirming that each primers pair is exclusively selective to its target gene.

#### 3.2.4. Evaluation of Sensitivity

##### Detection Limits of Simplex and mPCR, from Individual Cultures

To determine detection limits of each PCR simplex under the optimized conditions, several PCRs were carried out following a decreasing gradient of the DNA used in the reactions. DNA quantities used in each PCR mixes were calculated from a linear regression curve. This curve was established using the EXCEL software, from measurements made on concentrated dilutions of each DNA used (Figure 2, Figure 3, Figure 4 and Figure 5).

PCR detection limits, when the four primer pairs and their target DNAs were mixed in a single reaction, was evaluated by several PCR mixes containing the same components, whose DNA concentrations differed, following a decreasing gradient, with equal ratios between the four target microorganisms in the same reaction. DNA quantities used were calculated, following the same procedure previously mentioned.

The results showed a very clear detection of the four microorganisms (Figure 6), with intense bands (Lane 1). However, by reducing the DNA amount even more (Lane 2), all of the bands were still present, but intensity was lost, which means that the detection limits of the quadruplex PCR are approximately 10 pg/µL of each DNA template.

##### Detection Limits for Co-Culture in BPW

Aliquots of each co-culture were tested by mPCR, to evaluate the effect of BPW as a co-culture medium. The mPCR was able to detect simultaneously the four microorganisms, from the DNA extracted from co-cultures initially inoculated at 10^3^, 10^2^, 10^1^, and 10^0^ CFU/mL, even if the bands intensity clearly decreased for 10^1^ CFU/mL and 10^0^ CFU/mL initial inoculum, especially for *S. aureus* and *L. monocytogenes*. On the other hand, the intensity of *E. coli* and *S. enterica* bands was strong and invariable from 10^3^ to 10^0^ CFU/mL initial inoculum, with predominance of *E. coli* (Figure 7).

##### Detection Iimits from BPW Co-Culture from Artificially Inoculated Food Matrix

Multiplex PCR (mPCR) was applied to determine the efficiency of detection from BPW co-cultures in the presence of ready-to-eat food matrices artificially inoculated, with the presence of background microbiota.

The mPCR performed with the extracted DNA of co-culture from the eco-organic lettuce artificially inoculated with 10^3^ CFU/mL of each target bacteria was able to detect *E. coli*, *S. enterica* and *L. monocytogenes* very clearly, without any non-specific products (Figure 8a).

For the co-culture from meat sample artificially inoculated with 10^3^ CFU/mL of each target pathogen, after incubation mPCR was able to detect *E. coli*, *S. enterica* and *L. monocytogenes* (Figure 8a, Lane 3).

*S. aureus* could not be detected by mPCR from any of the aliquots tested (Lane 1, Lane 2 and Lane 3). However, it was clearly detected by simplex PCR from the co-culture aliquots of the two artificially inoculated samples (Figure 8b): from lettuce aliquot at 8.75 × 10^6^ CFU/mL, results presented in (Lane 4) and, more especially, from the meat samples, where a difference in the bands intensity was noticed, before incubation at 1.10 × 10^3^ CFU/mL (Lane 5) and after incubation at 4.45 × 10^7^ CFU/mL (Lane 6).

## 4. Discussion

This study started from the concept of using the simplest techniques, as well as the least expensive means possible, available in most of laboratories, to develop a rapid method for a simultaneous detection of the four microorganisms *E. coli*, *S. aureus*, *L. monocytogenes*, and *S. enterica* from raw and ready-to-eat foods. Our goal was to develop a protocol consisting of the combination of a 24 h co-culture step in a single broth combined with a standard mPCR detection.

The results obtained from efficiency comparison between BPW and the two non-selective liquid broths, LB and NB, in individual culture were globally very close, especially for BPW and NB, with a growth exceeding 9.00 log_10_ CFU/mL for *E coli* and *S. aureus*, and equal or above 8 log_10_ CFU/mL for *L. monocytogenes* and *S. enterica.* However, we noticed that the growth of the target bacteria in NB was slightly higher than the resulting from BPW, especially for *S. aureus* and *L. monocytogenes*, which may be linked to the presence of beef extract in NB composition. Abd El-Salam et al. (2010) [47] made the same observations about the growth of *S. enterica* in BPW and *L. monocytogenes* in NB broth.

Although NB showed better recovery rates for *L. monocytogenes* and *S. aureus*, the performance of both culture media was very good. Thus, we chose BPW medium since it is generally used as pre-enrichment broth in many ISO protocols, in addition to its high ability to elute the bacteria from leafy vegetables as lettuce [16]. Moreover, NB contains beef extract and some of its components, such as fats or muscle proteins, are considered as inhibitors that can affect PCR at multiple steps [48]. Wang and Suo (2011) [49] obtained very good detection results after 16 h of growth in BPW, using meat artificially contaminated by *E. coli* O157 and *Salmonella* Enteritidis, without any problematic interference due to the presence of background microbiota. The same observation was made by Alarcon et al. (2004) [50] for the simultaneous detection of *Salmonella* spp., *S. aureus* and *L. monocytogenes* from artificially inoculated samples.

The efficiency of BPW to support the growth of each target bacteria in individual culture was compared with other selective and non-selective media. Regarding the selective respective media (BGBLB, GC, RV, FB) used for the enrichment step in the ISO protocols of each target pathogen, BPW showed better performance for the growth of *E. coli*, *S. aureus* and *S. enterica* than BGBLB, GC, and RV, respectively. Similar results were observed for the growth of *S. enterica* serotype Enteritidis, in RV broth by Yu et al. (2010) [35].

The only selective medium that achieved better growth compared to BPW was Fraser medium, with a maximum density of *L. monocytogenes* at 9.94 log_10_ CFU/mL, compared to 8.00 log_10_ CFU/mL in BPW, which nevertheless remains more than satisfactory. The same observation was made in the work of Yu et al. (2010) [35] and Chen et al. (2015) [5].

Due to the promising results of the selective multi-pathogen enrichment broth SSSLE of Chen and al. (2015) [5], it was added to this study, in order to specifically enrich *E. coli*, *S. aureus*, *L. monocytogenes*, and *S. enterica*. And since this broth was formulated based on BPW composition, inhibitors for selectivity against food background microbiota and growth promoters, such as esculin for *L. monocytogenes* and mannitol for *S. aureus*, were added. Unfortunately, the only growth recorded during our test was *E. coli* at 5.72 log_10_ CFU/mL and no growth was observed for the other three target microorganisms.

To evaluate the recovering abilities, individual cultures, and co-culture in BPW broth were performed, with low initial inoculums of 10^3^, 10^2^ and 10^1^ CFU/mL. In individual culture, the maximum densities obtained after 24 h of incubation in BPW *E. coli* yielded the highest and the most stable recovery levels, with a maximum density of 8.93–8.71 log_10_ CFU/mL, closely followed by *S. aureus* 8.51–8.12 log_10_ CFU/mL and *S. enterica* 8.34–8.03 log_10_ CFU/mL. *L. monocytogenes* obtained the lowest maximum density, ranging between 7.32–7.18 log_10_ CFU/mL, which nevertheless remains widely detectable by most biomolecular tools.

Regarding the 24 h co-cultures in BPW broth, we found that the growths recorded for all target microorganisms were only slightly affected by the initial inoculum levels decrease. However, the competition effect significantly seemed to affect the resulting recovery rates. The competitive effect between the four microorganisms was demonstrated by the growth rates comparison of each target bacteria, obtained from individual cultures and co-culture under the same inoculation and growth conditions: Gram-negatives seemed to be the least affected, with an average decrease between 0.09–1.10 log_10_ CFU/mL, while Gram-positives showed a more significant decrease in co-culture, with an average decrease of 2.03–2.39 log_10_ CFU/mL. This competitive effect during simultaneous growth has been previously mentioned [5,51].

*E. coli* was not affected by the presence of other bacteria and its growth rates remained stable, while *S. enterica* showed a slight but insignificant decrease. Regarding *L. monocytogenes*, the growth rates recorded during co-culture were lower than in individual culture, which is in accordance with Daley et al. (2014) [52], who showed that the competitive effect exerted by *Enterobacteriaceae* could lead to a decrease in *L. monocytogenes* population ranging from 1 to 4 logs during 48 h of enrichment. Kim and Bhunia (2008) [34] suggested that the fast-growing *E. coli* and *Salmonellae* probably used up most of the nutrients and depleted the culture medium, resulting in a lower growth rate for *L. monocytogenes*, which is known to be a slow-growing bacteria and a poor competitor.

The maximum population rate in co-culture of *S. aureus* was not the lowest (6.23–5.62 log_10_ CFU/mL), but it was the most affected pathogen by the presence of other bacteria, showing the most important density decrease in co-culture, compared to its individual growth. This inhibition decreased with reduction of *E. coli* inoculum. Oberhofer and Frazier (1961) [53] reported that, although *E. coli* did not cause any obvious inhibition of *S. aureus* on spot plates, it could nevertheless strongly suppress growth of *S. aureus* in liquid medium.

Although the recovery of the four microorganisms in BPW co-culture seems less effective compared to individual culture rates, the resulting growth remain more than sufficient, for mPCR detection.

Recovery capacity of BPW from food matrices artificially inoculated at low initial concentrations (10^3^ CFU/mL) was evaluated, in the presence of background microbiota. Both lettuce and minced meat matrices tested were chosen as ready-to-eat food models, since they are usually consumed (lettuce) or can be consumed (minced meat) without cooking treatment. Lettuce, mainly contaminated by soil or irrigation water [16], and minced meat, mainly by handling [21], can be vectors of Shiga toxin-producing *E. coli* (STEC), *Salmonella* spp., *L. monocytogenes* and *S. aureus* [54,55].

Regarding the co-culture from lettuce, the recovery of target microorganisms from 10^3^ CFU/mL initially inoculated was greater than 8 Log_10_ CFU/mL for Gram-negative, thus dominating the Gram-positive, which showed a recovery greater than 6 Log_10_ CFU/mL. However, compared to the co-culture in medium, a slight increase in the cell density in the presence of lettuce was recorded, around 0.92 Log_10_ CFU/mL for *S. enterica*, 0.71 Log_10_ CFU/mL for *S. aureus*, as well as 0.45 Log_10_ CFU/mL for *L. monocytogenes*, and a decrease of 0.59 Log_10_ CFU/mL for *E. coli*. The same observation was made for the co-culture from minced meat compared to the co-culture in medium, where the cell concentrations recovered were more for *S. aureus* (+1.42 Log_10_ CFU/mL) and for *S. enterica* (+0.54 Log_10_ CFU/mL). For *L. monocytogenes,* difference was not as significant (+0.03 Log_10_ CFU/mL). As for the test with lettuce, the recovery of *E. coli* showed a decrease of 1.17 Log_10_ CFU/mL compared to the co-culture only in culture medium. This means there was no significant inhibitory effect of the matrix, as well as its background microbiota, during the co-culture in BPW. Therefore, the use of selective agents was not necessary. Moreover, the presence of foods seems to have enriched the co-culture, thus stimulating the growth of target bacteria.

The design of our mPCR assay was made by the combination of four pairs of primers developed in previous research, for simultaneous detection of *E. coli* [41], *S. aureus* [42], *L. monocytogenes* [43], and *S. enterica* [44], and they have been described as very specific and reliable primers to target the bacteria studied [3,11,16,56,57,58].

The specificity of the primer sequences was checked first in silico by the Primer-BLAST software (http://www.ncbi.nlm.nih.gov/tools/primer-blast/ access on 25 November 2021) [54], then in vitro by duplex, triplex and quadruplex PCR combinations. Multiplex PCR (mPCR) conditions such as annealing temperature, DNA template quantity and concentrations of the reaction mix components were optimized to simultaneously amplify the four target fragments, obtaining the highest detection sensitivity possible. One of the most critical points of this optimization was to find the balance between the concentrations of the four primers, to avoid the preferential amplification of certain target over another and ensure the most homogeneous amplification.

One of the essential factors affecting the PCR sensitivity is the establishment of a reliable DNA extraction procedure [20,59]. On this basis, we compared two widely used methods, namely, the extraction by Thermal lysis method (boiling method) without lysozyme, proposed by Zhang et al. (2012) [11], as well as the extraction by GenElute ™ bacterial genomic DNA extraction kit with lysozyme. After detection by simplex PCR, we were able to confirm that the extraction could be carried out correctly by thermal lysis, even without lysozyme. However, even if the boiling method costs nothing, saved time and eliminate the need of intensive labor [3,60], the DNA extraction by kit is much more efficient regarding the quantity and quality of DNA recovered, which is important during the detection of small amounts of DNA.

mPCR abilities were evaluated firstly, by the determination of its selectivity, which was confirmed by the absence of amplification for non-target bacteria tested. Then the detection sensitivity of each pair of primers was tested by simplex PCR, according to a DNA concentration gradient, from individual pure cultures in BPW. Using the same approach, the detection sensitivity of the mPCR was evaluated, by mixing the four pairs of primers with the four target DNAs, with approximately the same DNA ratio for each target bacteria, in the same reaction. The results obtained showed that reliable simultaneous amplification of the four target microorganisms (quadruplex PCR), under the optimized conditions could be observed until approximately 10 pg/µL of each DNA template in one reaction, which means that each primers pair, tested simultaneously or separately under these optimized conditions, was sensitive and specific enough to detect its target pathogen.

The mPCR detection carried out on co-cultures from the different BPW broth inoculations (10^3^, 10^2^, 10^1^, 10^0^ CFU/mL) was able to successfully amplify DNA fragments of *E. coli*, *S. aureus*, *L. monocytogenes* and *S. enterica* up to 10^0^ CFU/mL initial inoculum, which confirms the ability of BPW as a co-culture medium for mPCR-based detection. Difference in bands intensity during mPCR detection may be related to the difference in cell concentrations recovered for each pathogen [61]. However, according to Yuan et al. (2009) [57] the traditional mPCR cannot avoid disproportionate amplification between the different primers during the whole reaction, due to the fact that each primers pair have different amplifying efficiencies. On this point, Zhang et al. (2012) [11] and Wei et al. (2018) [3] who used the same primers, for the simultaneous detection of *S. aureus* and *Salmonella* spp., noticed that there was fierce competition between the different amplification pathways.

mPCR from co-culture in the presence of food matrices artificially inoculated with 10^3^ CFU/mL of each target microorganisms was able to detect very clearly the presence of *E. coli*, *S. enterica* and *L. monocytogenes*, despite the background microbiota, after incubation. The difference of bands intensity between the two food matrices could be due to the difference in recovered cell concentrations, which were more important from the lettuce, but also to a possible inhibitory effect of meat. According to Garrido et al. (2013) [62] and Rajabzadeh et al. (2018) [16], fats and glycogen are considered as inhibitors and could affect PCR at multiple steps.

Unfortunately, in the presence of food, *S. aureus* could be detected only by simplex PCR from co-culture. Nevertheless, we noticed an increase of detection intensity after incubation in BPW, compared to the detection from initial inoculum. We assume that the problem is related to the sensitivity of the primers used, and that they should be replaced by more efficient ones. Wei et al. (2018) [3] indicated that detection of *S. aureus* with Nuc484 primers was not characterized by high sensitivity compared to *Salmonella* spp. primers invA284 in artificially contaminated food products. Zhang et al. (2012) [11] confirmed that the biomass of *S. aureus* had to be greater than other bacteria for a perfect amplification of all of the target genes, due to a possible competition established by a preferential amplification. According to Elizaquível and Aznar (2008) [10], the preferential amplification of a target in mPCR can be induced by the presence of a low template concentration. In our case, the template-primer combinations of *E. coli*, *S. enterica*., and *L. monocytogenes*, seems to be more favored, thus affecting the amplification efficiency for *S. aureus* [63].

## 5. Conclusions

In conclusion, BPW broth can effectively support the individual and simultaneous growth of *E. coli*, *S. aureus*, *L. monocytogenes*, and *S. enterica* from low initial contamination levels in the presence of ready-to-eat food matrix such as eco-organic lettuce or minced meat, and could be, thus, used as a co-culture medium prior to mPCR detection. DNA extraction by thermal lysis method was effective on both Gram-negative and Gram-positive bacteria, even without lysozyme. However, the specific DNA extraction kit used in this work can improve the efficiency of the mPCR assay. Although mPCR could not detect *S. aureus* in artificially inoculated food matrices, the proposed protocol, which includes a BPW pre-enrichment step and detection by mPCR, can confirm the presence or absence of *E. coli*, *Salmonella* sp. and *L. monocytogenes* in foods in approximately 30 h, compared to cultural methods which require at least seven days, therefore considerably reducing the time, efforts, and costs of analyzes.

## Figures and Tables

**Figure 1 ijerph-19-01031-f001:**
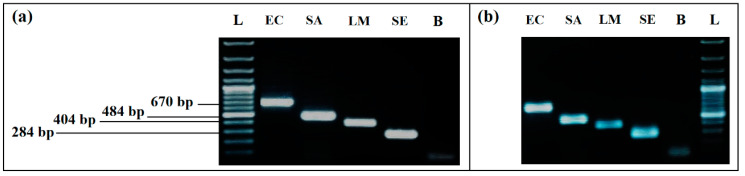
Comparison of the two DNA extraction methods effect on PCR detection: (**a**) simplex PCR with DNA extracted by GenElute™ bacterial genomic DNA Extraction kit, with lysozyme. (**b**) simplex PCR with DNA extracted by thermal lysis protocol, without lysozyme. L, 100 bp DNA ladder; EC, *E. coli* (8.55 × 10^8^ CFU/mL); SA, *S. aureus* (3.27 × 10^8^ CFU/mL); LM, *L. monocytogenes* (2.1 × 10^7^ CFU/mL); SE, *S. enterica* (2.2 × 10^8^ CFU/mL); L, 100 bp DNA ladder; B, negative control.

**Figure 2 ijerph-19-01031-f002:**
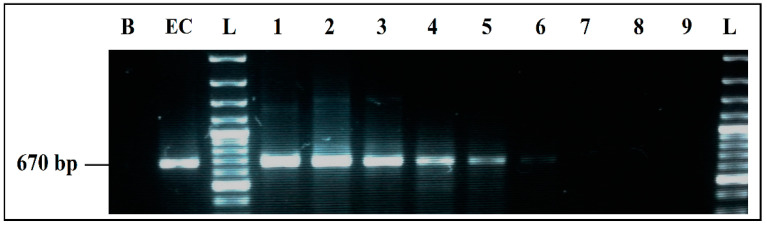
Detection sensitivity of *E. coli* primers GADA 670 bp in simplex PCR. B, negative control; EC, positive control *E. coli* 101 CECT; L, 100 bp DNA ladder; lane 1 to 6, PCR results from 6078 pg/µL to 0.06 pg/µL DNA; Lane 7 to 9, PCR results from 0.006 pg/µL to 0.00006 pg/µL DNA; L, 100 bp DNA ladder.

**Figure 3 ijerph-19-01031-f003:**
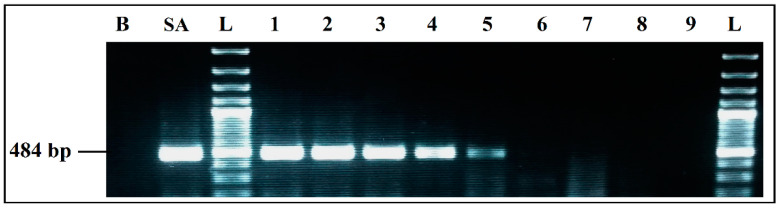
Detection sensitivity of *S. aureus* primers Nuc 484 bp, in simplex PCR. B, negative control; SA, positive control *S. aureus* 435 CECT; L, 100 bp DNA ladder; lane 1 to 6, PCR results from 2457 pg/µL to 0.03 pg/µL DNA; lane 7 to 9, PCR results from 0.003 pg/µL to 0.00003 pg/µL DNA; L, 100 bp DNA ladder.

**Figure 4 ijerph-19-01031-f004:**
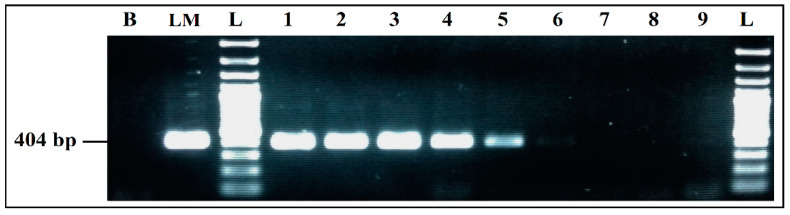
Detection sensitivity of *L. monocytogenes* primers LM 404 bp, in Simplex PCR. B, negative control; LM, positive control *L. monocytogenes* 936 CECT; L, 100 bp DNA ladder; lane 1 to 6, PCR results from 359 pg/µL to 0.004 pg/µL DNA; lane 7 to 9, PCR results from 0.0004 pg/µL to 0.000004 pg/µL DNA; L, 100 bp DNA ladder.

**Figure 5 ijerph-19-01031-f005:**
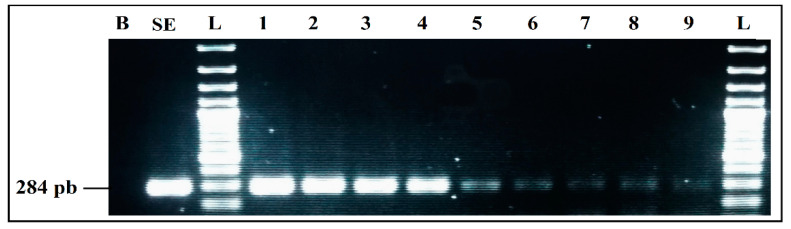
Detection sensitivity of *S. enterica* primers SalinA 284 bp, in Simplex PCR. B, negative control; SE, positive control *S. enterica* 4266 CECT; L, 100 bp DNA ladder; lane 1 to 9, PCR results from 4547 pg/µL to 0.00005 pg/µL DNA; L, 100 bp DNA ladder.

**Figure 6 ijerph-19-01031-f006:**
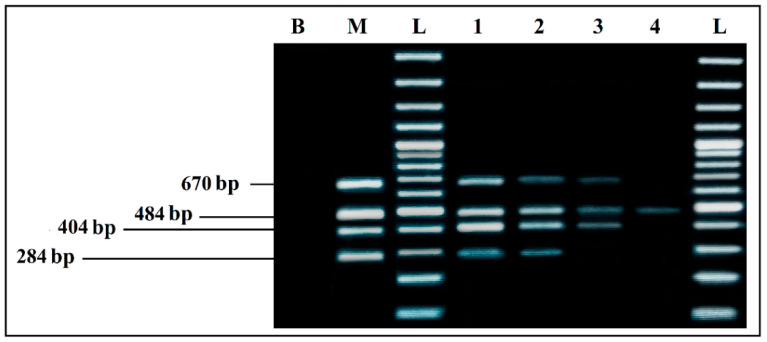
Detection sensitivity of mPCR from individual cultures. B, negative control; M, mPCR positive control; L, 100 bp DNA ladder; lane 1, mPCR detection with mix DNA (100–70 pg/µL) for each target; lane 2, mPCR detection with mix DNA (10–8 pg/µL) for each target; lane 3, mPCR detection with mix DNA (5–1 pg/µL) for each target; lane 4, mPCR detection with mix DNA (0.6–0.1 pg/µL) for each target; L, 100 bp DNA ladder.

**Figure 7 ijerph-19-01031-f007:**
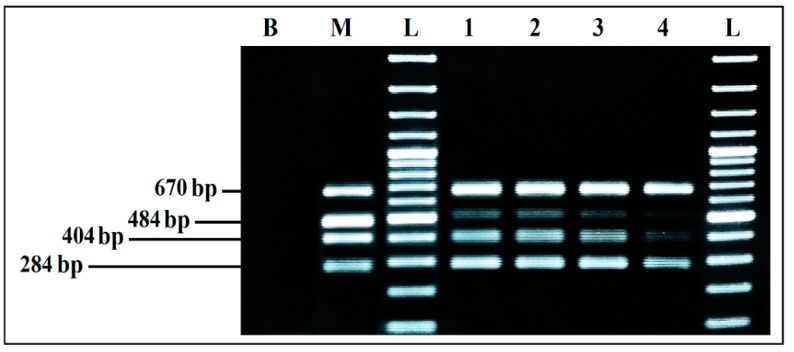
mPCR detection, from co-cultures in BPW. B, negative control; M, mPCR positive control; L, 100 bp DNA ladder; lane 1, mPCR results from co-culture in BPW initially inoculated at 10^3^ CFU/mL; lane 2, mPCR results from co-culture in BPW initially inoculated at 10^2^ CFU/mL; lane 3, mPCR results from co-culture in BPW initially inoculated at 10^1^ CFU/mL; lane 4, mPCR results from co-culture in BPW initially inoculated at 10^0^ CFU/mL; L, 100 bp DNA ladder.

**Figure 8 ijerph-19-01031-f008:**
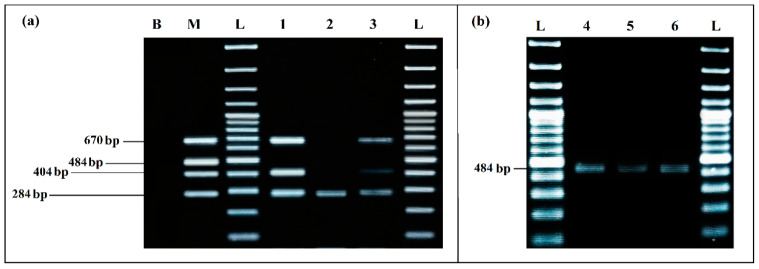
PCR detection from food matrices artificially inoculated and enriched in BPW: (**a**) mPCR. B, negative control; M, positive control; L, 100 bp DNA ladder; lane 1, mPCR results from lettuce artificially inoculated (10^3^ CFU/mL); lane 2, mPCR results from minced meat artificially inoculated (10^3^ CFU/mL); lane 3, mPCR results from minced meat artificially inoculated (10^3^ CFU/mL); L, 100 bp DNA ladder. (**b**) Simplex PCR for *S. aureus*; L, 100 bp DNA ladder; lane 4, PCR results from lettuce artificially inoculated (10^3^ CFU/mL); lane 5, PCR results from minced meat artificially inoculated (10^3^ CFU/mL) before incubation; lane 6, PCR results from minced meat artificially inoculated (10^3^ CFU/mL) after incubation; L, 100 bp DNA ladder.

**Table 1 ijerph-19-01031-t001:** Primers used.

Strains	Primers	Sequences	Size	References
** *E. coli* **	GADA/F	ACCTGCGTTGCGTAAATA	670 bp	McDaniels et al., 1996 [41]
	GADA/R	GGGCGGGAGAAGTTGATG		
** *S. aureus* **	Nuc/F	CTTTAGCCAAGCCTTGACGAAC	484 pb	Xu et al., 2006 [42]
	Nuc/R	AAAGGGCAATACGCAAAGAGGT		
** *L. monocytogenes* **	LM404/F	ATCATCGACGGCAACCTCGGAGAC	404 bp	Wu et al., 2004 [43]
	LM404/R	CACCATTCCCAAGCTAAACCAGTGC		
** *S. enterica* **	SalinvA139	GTGAAATTATCGCCACGTTCGGGCAA	284 bp	Rahn et al., 1992 [43]
	SalinvA141	TCATCGCACCGTCAAAGGAACC		

**Table 2 ijerph-19-01031-t002:** Individual culture growth (CFU/mL).

Bacteria	Selective Enrichment (BGBLB, RV, GC, FB)	NB	BPW	LB	SSSLE
*E. coli*	9.30 × 10^7^	1.48 × 10^9^	1.19 × 10^9^	2.73 × 10^8^	5.20 × 10^5^
*S. enterica*	6.83 × 10^7^	3.29 × 10^8^	4.10 × 10^8^	4.26 × 10^7^	0
*S. aureus*	1.74 × 10^8^	7.70 × 10^10^	1.07 × 10^9^	7.88 × 10^8^	0
*L. monocytogenes*	8.65 × 10^9^	8.15 × 10^8^	9.97 × 10^7^	3.49 × 10^8^	0

**Table 3 ijerph-19-01031-t003:** Recovery rates values in CFU/mL from individual and co-culture growths in BPW.

Bacteria	Initial Inoculum	Individual Culture	Co-Culture
*E. coli*	10^3^ CFU/mL	8.55 × 10^8^	7.70 × 10^8^
*S. aureus*		3.27 × 10^8^	1.70 × 10^6^
*L. monocytogenes*		2.10 × 10^7^	4.60 × 10^5^
*S. enterica*		2.20 × 10^8^	1.60 × 10^7^
*E. coli*	10^2^ CFU/mL	7.50 × 10^8^	5.65 × 10^8^
*S. aureus*		4.60 × 10^8^	2.70 × 10^5^
*L. monocytogenes*		1.75 × 10^7^	1.38 × 10^5^
*S. enterica*		1.28 × 10^8^	9.95 × 10^6^
*E. coli*	10^1^ CFU/mL	5.15 × 10^8^	4.20 × 10^8^
*S. aureus*		1.32 × 10^8^	4.20 × 10^5^
*L. monocytogenes*		1.50 × 10^7^	6.85 × 10^4^
*S. enterica*		1.08 × 10^8^	9.75 × 10^6^

**Table 4 ijerph-19-01031-t004:** Recovery rates values of co-culture growth in CFU/mL from BPW with and without food matrices initially inoculated at 10^3^ CFU/mL.

Bacteria	Without Matrix	With Lettuce Matrix	With Minced Meat Matrix
*E. coli*	7.70 × 10^8^	2.00 × 10^8^	5.20 × 10^7^
*S. aureus*	1.70 × 10^6^	8.75 × 10^6^	4.45 × 10^7^
*L. monocytogenes*	4.60 × 10^5^	1.30 × 10^6^	4.85 × 10^5^
*S. enterica*	1.60 × 10^7^	1.32 × 10^8^	5.50 × 10^7^

## Data Availability

Not applicable.

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
