# Peer review of "Simultaneous Detection of Four Main Foodborne Pathogens in Ready-to-Eat Food by Using a Simple and Rapid Multiplex PCR (mPCR) Assay"

_ijerph, 2022, doi:10.3390/ijerph19031031_

Round 1

Reviewer 1 Report

n this review manuscript, the authors reported a simultaneous detection of E. coli, L. monocytogenes, S. aureus and S. enterica based on an enrichment step in a single co-culture broth and a multiplex PCR assay in a short time while the sensitivity and the selectivity are both good enough.I think this paper can be acceptable after the following points are clarified and modified.

1.In Table 1,the covery performance for the four target bacteria in NB were much higher than growth in BPW except S. enterica, why not choose NB as co-culture medium.

2.In preparation of DNA template, the extraction by Thermal lysis method without lysozyme and GenElute ™ bacterial genomic DNA Extraction kit with lysozyme were selected, which method is better is not clearly described in the result part. When evaluating the detection method, the method used to extract the DNA of the sample is not clearly described, which should be explained.

Reviewer 2 Report

The manuscript title ” Co-culture and detection of E. coli, L. monocytogenes, S. aureus and S. enterica by multiplex PCR” is valuable but need a little more work. The calculations about sensitivity and specificity of the proposed multiplex PCR (http://www.winepi.net/uk/index.htm or any difference program https://epitools.ausvet.com.au/testevaluation) is necessary and it is my major comments, of course if it is possible. If it will be work Congrats to the Authors because it will be very useful. In all text is the problem with the abbreviation. 

Minor comments:

Line 2; Title, In my opinion in the Title should be the whole name of bacteria

Line 11; 13; Abstract, whole name and abbreviation “PCR, E.coli, L. monocytogenes, S. aureus and S. enterica

Line 15: “Buffered Peptone Water (BPW)”

Line 35; An abbreviation after the whole name “FDOSS”

Line 41; In “Introduction” E. coli for the first time for this an abbreviation after the whole name

Line 57-58; “Listeria monocytogenes, Staphylococcus aureus, Salmonella enterica and E. coli” an abbreviation after the whole name

Line 84; “multiplex PCR (mPCR)”

Line 90-97; For the first time the whole name of bacteria and after abbr.

Line 106-109; “mPCR”

Line 111; “Materials and Methods 2.1. Bacterial strains” problem with abbr.

Line 161, 165, 183; “Buffered Peptone Water” only abbr. because the Authors described this in line 132 in this paragraph

Line 207; “2.3.2. Primers” In my opinion, in this part, the table with sequence primer will be useful. The article is about new methods and will be nice to have all dates to do this.

Line 218; the references for BLAST

Line 240-242; a problem with abbr.

Line 247-251; A little chaos, “15 µL of DNA” In my opinion, the DNA should be put in ng. and the primers are in final concentration and it is good but this 15 µL of DNA on 29 µL of total volume something should be changed

Line 268-270; a problem with abbr.

Line 272-275; a problem with abbr. and Italic

Line 278; mPCR

Line 361-367; problem with abbr.

Line 461; “Escherichia coli” abbr.

Line 576- 588; The Authors should decide if they use abbr. mPCR or whole name multiplex PCR

Line 628; “References should be prepared according to Instructions”

Author Response

Thank you

Reviewer 3 Report

COMMENTS TO THE AUTHOR:

Co-culture and detection of E. coli, L. monocytogenes, S. aureus and S. enterica by multiplex PCR

Comments;

Overall, the entire manuscript is written in an appropriate English level but it needs some minor revisions as follows;

Comments to the Author
The manuscript entitled: " Co-culture and detection of E. coli, L. monocytogenes, S. aureus and S. enterica by multiplex PCR " is in the field of food Biotechnology and the results are very important for the safety of the community. Research has enough novelty for publication and needs some minor improvement as follow:
1- Title: If you can rewrite and make it more interesting for readers.

2- Abstract: It is OK

3-1- Keywords: Please choose keywords other than the main words of the title. In this case, other researchers can find your article by searching a wide range of words through databases.

3-2- Please provide Abbreviation section consequent the Keywords

4- Introduction was better shortened as a concise text.

5- Methodology: Please use the appropriate references for each method. For example Please cite

  • https://doi.org/10.1007/s11250-012-0123-3 for “3.4. Multiplex PCR” section
  • https://doi.org/10.5897/AJMR12.068               for “ introduction” section
  •  

6- Please double check some types in the whole manuscript
7- Results and discussion are OK.
8- Conclusion is very short, try to make it more comprehensive and concise.

Author Response

Thank you

Round 2

Reviewer 2 Report

The manuscript title the “Simultaneous detection of four main foodborne pathogens in ready-to-eat food by using a simple and rapid multiplex PCR (mPCR) assay” is now well prepared and the title is  good. As my major comments:

“The calculations about sensitivity and specificity of the proposed multiplex PCR (http://www.winepi.net/uk/index.htm or any difference program https://epitools.ausvet.com.au/testevaluation) is necessary and it is my major comments, of course, if it is possible.” I think that it was impossible but as a teacher who learns students about epidemiology, I have to ask about these calculations. Now I have only minor comments:

Minor comments:

Line 15; After a whole name of bacteria put abbr. “Escherichia coli (E. coli), Listeria monocytogenes (L. monocytogenes), Staphylococcus aureus (S. aureus) and Salmonella enterica (S. enterica)

Line 23; “Buffered Peptone Water” BPW, because line 18

Line 136; “Citrobacter freundii CECT 401, Micrococcus luteus  CECT 245, Staphylococcus epidermidis CECT 231 and 3 laboratory isolates (Listeria innocua, Listeria grayi and Bacillus cereus) put an abbr. after bacteria name, because the Authors used them in lines 410-413

Line 231; “Multiplex PCR (mPCR)“

Line 270; “Multiplex PCR (mPCR)“

Line 271: mPCR

Line 313; “multiplex PCR” mPCR

Line 320; 330 “mPCR”

Line 397; “mPCR” ???

Line 415; mPCR

Line 451; mPCR

Line 464; mPCR

Line 480; mPCR

Line 504; mPCR

Line 633; mPCR

Line 650, 674; mPCR

Line 697; “Buffered Peptone Water” BPW

Author Response

Dear Sirs:

Thank you again for your review and your very kind comment and suggestions. We have modified the text according to them. Main changes are highlighted in green colour:

Minor comments:

Line 15; After a whole name of bacteria put abbr. “Escherichia coli (E. coli), Listeria monocytogenes (L. monocytogenes), Staphylococcus aureus (S. aureus) and Salmonella enterica (S. enterica). Done

Line 23; “Buffered Peptone Water” BPW, because line 18. Done

Line 136; “Citrobacter freundii CECT 401, Micrococcus luteus  CECT 245, Staphylococcus epidermidis CECT 231 and 3 laboratory isolates (Listeria innocuaListeria grayi and Bacillus cereus) put an abbr. after bacteria name, because the Authors used them in lines 410-413. Done

Line 231; “Multiplex PCR (mPCR)“. Done

Line 270; “Multiplex PCR (mPCR)“. Done

Line 271: mPCR. Done

Line 313; “multiplex PCR” mPCR. Done

Line 320; 330 “mPCR”. Done

Line 397; “mPCR” ??? Done

Line 415; mPCR. Done

Line 451; mPCR. Done

Line 464; mPCR. Done

Line 480; mPCR. Done

Line 504; mPCR. Done

Line 633; mPCR. Done

Line 650, 674; mPCR. Done

Line 697; “Buffered Peptone Water” BPW. Done